# State Anxiety and Procrastination: The Moderating Role of Neuroendocrine Factors

**DOI:** 10.3390/bs13030204

**Published:** 2023-02-27

**Authors:** Efrat Barel, Shosh Shahrabani, Lila Mahagna, Refaat Massalha, Raul Colodner, Orna Tzischinsky

**Affiliations:** 1Department of Behavioral Sciences, The Max Stern Academic College of Emek Yezreel, Emek Yezreel 1930000, Israel; 2Department of Economics and Management, The Max Stern Academic College of Emek Yezreel, Emek Yezreel 1930000, Israel; 3Endocrinology Laboratory, Emek Medical Center, Afula 1855701, Israel

**Keywords:** procrastination, state anxiety, testosterone, cortisol

## Abstract

Procrastination is prevalent among students, as well as the general population, and has negative impacts on various domains. Several models aimed to understand factors associated with procrastination, with some suggesting that anxiety plays a significant role. Biological factors have been shown to contribute to individual differences in procrastination; however, little attention has been paid to the role of neuroendocrine factors on procrastination. The primary question addressed in the present study is whether neuroendocrine factors (testosterone and cortisol) moderate the association between state anxiety and procrastination. Eighty-eight participants (29 men; 32 women using oral contraceptives; and 27 women not using oral contraceptives and in their luteal phase) were tested for biomarkers and completed questionnaires. Results show that state anxiety is positively correlated with procrastination. Furthermore, testosterone levels moderate the correlation between state anxiety and procrastination. As testosterone levels drop, the positive correlation between state anxiety and procrastination becomes stronger, but when testosterone levels are higher, no significant association between state anxiety and procrastination is found. Cortisol levels do not moderate the relationship between state anxiety and procrastination. The role of neuroendocrine factors for psychological outcomes is discussed.

## 1. Introduction

Procrastination has been defined as a tendency to delay a task or decision [1], and incidence rates have increased in recent years. It is widespread in academic life and also in the general population, chronically affecting some 15–20% of adults [2]. Procrastination has been described and studied as an extensive and potentially harmful phenomenon in various arenas, such as health, economics, and politics [2]. For example, procrastinators exhibit more symptoms of physical illness and stress [3] and greater reluctance to engage in health-promoting behaviors [4]; procrastinators tend to lack retirement savings [5]; and in politics, procrastinators are characterized by delaying important decisions [6]. Given the extensive potential consequences of procrastination, a considerable amount of empirical work has been conducted in the past few decades to investigate the causes of and correlations to procrastination.

Several models have been proposed to uncover the factors influencing procrastination. Trait-based models have suggested that personality traits such as impulsiveness and conscientiousness are associated with procrastination [2]. Elaborations of these trait-based models generated models incorporating personality variables as well as cognitive, affective, and behavioral factors (e.g., [7]). In his integrative motivational model, Steel [2] posited that motivational effects (such as temporal distance to the task deadline) are moderated by personality factors (such as impulsivity and low self-discipline) [8].

Over the past three decades, a great deal of attention has been drawn to the relationship between anxiety and procrastination. A few studies investigated the role of anxiety in individual differences in procrastination and suggested that procrastination may elevate unpleasant feelings such as stress and anxiety [9]. However, most studies regarded anxiety as the antecedent to procrastination [10]. While some studies focused on trait anxiety and others on state anxiety or domain-specific anxiety, in most studies, anxiety has been consistently found as positively correlated with procrastination [10,11]. This association was not, however, documented by all studies: several found a positive correlation between trait anxiety and procrastination [12], while others found a correlation between domain-specific anxiety and procrastination but with the sign of the relationship depending on tasks [13].

Overall, studies have shown varying degrees of association between anxiety and procrastination with a range of possible explanations. Some researchers asserted that, according to clinical observations, fear of failure is the cause of increased procrastination [14]. Others suggested that individuals look to escape a task in order to relieve negative feelings such as anxiety [15]. On the other hand, it has also been posited that, under certain conditions, anxiety might reduce procrastination. For example, Steel [2] suggested that the relationship between anxiety and procrastination is moderated by impulsivity.

The linkage between procrastination and negative mental health states, such as anxiety, has prompted investigations focusing on physiology. For example, a few studies explored the relationship between procrastination and physiological measures of stress. Stress is regulated by the sympathetic nervous system (SNS) and the hypothalamic–pituitary–adrenal (HPA) axis. Stress triggers the SNS to release catecholamines with salivary alpha amylase (sAA), a digestive enzyme found in the oral cavity that serves as a marker for SNS [16]. The main end product of the HPA axis in humans is cortisol (C) [17]. Khalid and colleagues [18] found that participants with higher procrastination had higher levels of perceived stress and sAA. Furthermore, the C response to stress was found to be related to the emotion regulation difficulties associated with maladaptive perfectionism [19]. A meta-analysis reveals that procrastination is positively associated with this maladaptive form of perfectionism [20], thus, suggesting a linkage between C and procrastination.

Gonadal hormones have been also studied in relation to emotion regulation and stress reactivity. Testosterone (T), in particular, has been shown to modulate the HPA axis and to be negatively correlated with cortisol reactivity to psychosocial stress [21]. A considerable body of research focused on the effects of T and C on various social behaviors, such as dominance, aggression, and competition. Most studies investigated T and C independently, but in the last two decades, various neurobiological studies focused on their potential joint regulation of social behavior. The results are inconsistent, with some showing that elevated T levels and reduced C levels are associated with aggressive and dominant behaviors [22,23] while others have failed to demonstrate this association [24,25].

There is little research on the involvement of neuroendocrine measures in procrastination; however, recently, Jamieson and colleagues [26] applied a stress regulation technique to assess psychophysiological outcomes in evaluative academic contexts. Specifically, they tested the effects of stress reappraisal (the acquired information about the functional benefits of stress) on psychological, biological, and performance outcomes. They randomly assigned community college students to stress reappraisal and control conditions before taking an exam. They found that reappraising stress led to less math anxiety, lower C levels, higher T levels, and less procrastination, as well as other beneficial outcomes. This study demonstrates the interconnectivity between neuroendocrine and psychological factors and highlights the need to incorporate biological measures to deepen our understanding of individual differences in psychological outcomes such as procrastination.

The present study aimed to expand our understanding of the relationship between anxiety and procrastination through neuroendocrine measures. Specifically, we hypothesized that state anxiety would be positively associated with procrastination. We tested state anxiety rather than trait anxiety due to previous suggestions that the relationship between anxiety and procrastination may be bidirectional [10]. In addition, given the association recently found between anxiety, neuroendocrine measures, and procrastination [26], and based on previous studies regarding the role of T and C in various social behaviors, we tested the moderating effect of T and C on the association between anxiety and procrastination. T levels have been found to differ between men and women. Moreover, oral contraceptives have been seen to reduce androgen levels [27], while studies have shown the effects of oral contractive use on cortisol reactivity to induced stress [28]. Due to these earlier findings, we included three study groups: men, natural cycling women, and women using oral contraceptives. We hypothesized that C would moderate the relationship between anxiety and procrastination: specifically, higher levels of C would strengthen the positive association between anxiety and procrastination. Lastly, we hypothesized that T would moderate the relationship between anxiety and procrastination: specifically, lower levels of T would strengthen the positive association between anxiety and procrastination.

## 2. Materials and Methods

### 2.1. Participants

The participants were 88 undergraduate students (29 men). Of the women, 32 were taking oral contraceptives (oral contraceptive group; OC), and the remaining 27 were not taking oral contraceptives and were in their mid-luteal phase (luteal phase group; LP). Participants were recruited through advertisements on college notice boards. The exclusion criteria included: individuals with serious medical, gynecological, or hormonal problems; smokers; and individuals with self-reported attention deficit hyperactivity disorder (ADHD) or learning disabilities. All the male participants met the inclusion criteria. The women participants were, in addition, pre-screened to verify meeting the inclusion criteria. Women to be included in the OC group were all tested during the on-phase of pill intake and were using pills containing 25 mg of estrogen (ethinylestradiol) and 75 mg of progestin (gestodene). These doses are considered moderate and are commonly prescribed, and they are considered as having antiandrogenic properties [29]. All women had been taking the pills for at least one year. The women included in the LP group had not been taking oral contraceptives for at least six months prior to the study, had a regular menstrual cycle, and were not pregnant or lactating. These participants were monitored for at least three months prior to the study in order to verify the regularity of their cycles and were summoned to the research laboratory on the 21st day of their cycle using the day of the onset of their last menstruation as a reference point [30]. The sample had a mean age of 24.86 years (*SD* = 2.93) and a mean BMI of 23.38 (*SD* = 3.73).

### 2.2. Measures

The study used self-report questionnaires to measure the following:

*Procrastination*. Lay’s General Procrastination Scale (GPS) [31] is a widely used and well-validated measure of procrastination across a range of tasks. GPS-9 is the short form composed of 9 items developed and validated by Sirois and colleagues [32]. It demonstrated good test–retest reliability, internal consistency, and construct validity across samples. The items are scored on a 5-point Likert-type scale ranging from 1 (disagree strongly) to 5 (agree strongly). The items were averaged to create an index of procrastination (Cronbach’s α = 0.88).

*State Anxiety*. The State-Trait Anxiety Inventory from Y-1 (STAI-S) [33] is a widely used and well-validated measure that assesses the extent of current feelings of anxiety. The 20 items are scored from 1 (not at all) to 4 (very much so). The STAI has demonstrated very good internal consistency and good item characteristics [34], and previous studies provided normative data for various age groups and countries (e.g., [34,35,36]). The items were summed and averaged to create an index of state anxiety (Cronbach’s α = 0.93).

*Personal details*. Socio-demographic information, including age, gender, weight, and height.

#### Biomarkers Measures

In order to test endogenous hormone levels, C and T levels were measured. In order to control diurnal rhythm changes in C and T, participants were tested between 12:00 and 18:00 pm. They were asked to refrain from eating, drinking (except water), and smoking for at least one hour prior arrival in the laboratory. Prior to the saliva sampling, participants were instructed to chew on a piece of parafilm for several seconds to increase saliva secretion. They then deposited a 2.5 mL of saliva in a SaliCap sampling vial (IBL International GMBH, Hamburg, Germany). Saliva samples were stored at −20 °C upon collection and until the laboratory tests were performed. For each biochemical analyte, tests were performed using commercial CE-IVD-approved ELISA kits: cortisol saliva ELISA (mean intra-assay CV% = 0.47, mean inter-assay CV% = 10.6, assay sensitivity = 0.014 nmol/L), testosterone saliva ELISA (mean intra-assay CV% = 5.57, mean inter-assay CV% = 5.7, assay sensitivity = 8.6 pg/mL) (all from IBL International GMBH, Hamburg, Germany). All tests were performed in an SQII ELISA processor (AESKU Systems, Wendelsheim, Germany). All tests were performed in the endocrinology laboratory of Emek Medical Center, an ISO 9001 (2015 version)-certified and JCI-accredited facility. All analytical kits used in the study were previously validated in the laboratory according to good laboratory practice standards. A calibration curve using standard duplicates was performed for each analyte in every run.

### 2.3. Procedure

The study was approved by the institutional ethical review board (IRB) of the Max Stern Yezreel Valley College. After giving their fully informed consent, participants completed a brief demographic questionnaire. All participants were then tested for biomarkers, after which they completed the questionnaires.

#### Statistical Analysis

The statistics analyses were conducted using IBM SPSS Statistics 28.0 software. Due to significant differences in mean T between men, OC, and LP groups and previous suggestions regarding the possible association between C and T (e.g., [21]), C and T were standardized separately for men, OC, and LP groups. To test the moderation hypotheses, moderated regression analyses were conducted. Significant interactions were decomposed using the procedures described by Aiken and West [37]. To interpret significant interaction, we used the multiple regression model to plot procrastination scores one standard deviation below (low) and above (high) the means of T and anxiety.

## 3. Results

Table 1 shows the mean hormone concentrations of C and T for men, OC, and LP women. As expected, men’s T levels are higher than those of women. Consistent with previous research (e.g., [38]), the groups do not differ in C levels. While there was a marginally significant difference between the groups in state anxiety, post hoc analysis did not detect a significant effect. There is no significant difference between the groups in procrastination levels.

Table 2 shows zero-order correlations between hormones, procrastination, and anxiety. Procrastination is positively correlated with anxiety, while procrastination and anxiety are not significantly correlated with T or C levels.

A multiple regression was performed with procrastination scores as the dependent variable, anxiety as the independent variable, T as a moderator, T×anxiety interaction, and sex (dummy) as a covariate. The model explains 19% of the variance of procrastination (*F*(4,82) = 4.91, *p* < 0.01). The results reveal a significant T×anxiety interaction (*β* = −0.49, *p* < 0.05, 95% CI = −0.95, −0.03) (see Figure 1). Simple slope analyses [39] find that the lower the T levels, the stronger the association between anxiety and procrastination (for low levels of T: *b* = 1.24, *t*(86) = 4.38, *p* < 0.001; for medium levels of T: *b* = 0.77, *t*(86) = 3.89, *p* < 0.001), while for high levels of T, there is no significant association between anxiety and procrastination (*b* = 0.97, *t*(86) = 0.66, *p* > 0.05). A multiple regression was performed with procrastination scores as the dependent variable, anxiety as the independent variable, C as a moderator, C×anxiety interaction, and sex (dummy) as a covariate. The model explains 0.18% of the variance of procrastination (*F*(4,81) = 4.50, *p* < 0.01). However, there is no significant C×anxiety interaction (*β* = 0.49, *p* > 0.05, 95% CI = −0.18, 1.17).

## 4. Discussion

Consistent with previous studies, the total effect reveals that anxiety is correlated with procrastination. Higher levels of anxiety are associated with higher levels of procrastination. Indeed, previous studies demonstrate that anxiety, whether conceptualized as a personality trait or situational state, is associated with procrastination [11,40]. For example, Fritzsche and colleagues [41] examined the relationship between academic procrastination and performance and found that trait anxiety and state anxiety were both positively correlated with procrastination. Other studies found procrastination to be associated with other unwanted internal experiences: for example, fear of negative evaluation [42] and fear of failure [43]. It has been suggested that individuals may procrastinate to avoid these unwanted experiences, such as fear and anxiety, when confronting aversive tasks [44,45].

The present study’s main aim was to explore neuroendocrine variables as potential moderators of the association between anxiety and procrastination. We found that T levels moderate the correlation between anxiety and procrastination: the lower the T levels, the stronger the positive correlation between anxiety and procrastination. By contrast, when T levels are higher, no significant association between anxiety and procrastination is found. These findings suggest that T modulates psychological outcomes. Previous studies demonstrated the ability of T to reduce anxiety. For example, in a series of studies, Aikey and colleagues [46] show that the anxiolytic effects of T in mice are attributable to the influence of androgenic metabolites on the γ-aminobutyric acid (GABA_A_) receptor (an inhibitory neurotransmitter). The authors stated that reduction in anxiety may be relevant to reproductive success. In the present study, the relationship between anxiety and procrastination is diminished for higher T levels, thus, suggesting that the association between anxiety and procrastination is modulated by T levels.

The involvement of T levels in psychological outcomes calls for the examination of sex differences in procrastination. Due to sex differences in T levels, T levels were standardized within groups and sex was added to the regression as a covariate. Previous attempts to investigate sex differences in procrastination yielded mixed findings, with some showing that men exhibit higher levels of procrastination than women (e.g., [47]), whereas others did not find significant differences (e.g., [48]). In the present study no group differences in procrastination are found, suggesting that the moderation effect of T levels on the relationship between anxiety and procrastination does not differ between men and women.

In a recent experimental study, Jamieson and colleagues [26] revealed an association between test anxiety, procrastination, and neuroendocrine responses. Participants were randomly assigned to one of two experimental conditions—stress reappraisal and control condition—before taking an exam. Reappraising stress led to less test anxiety, less procrastination, higher T levels, and lower C levels. A psychological technique to enact positive change when facing stressful situations led to both neuroendocrine and psychological outcomes. Our findings partially support these findings by demonstrating a linkage between a neuroendocrine measure (T) and psychological outcomes. Nevertheless, our study did not detect a moderating effect of C levels. Our cross-sectional study measured neuroendocrine basal levels to explore individual differences in biological as well as psychological measures and their interconnectivity. Jamieson and colleagues’ study, on the other hand, was an experimental field study that manipulated conditions in order to optimize psychophysiological responses in stressful situations. Various studies investigated the joint influence of stress-induced activation of the HPA axis and the SNS and gonadal hormones on various psychological outcomes through the activation of the stress system. For example, gonadal hormones have been shown to modulate the effects of a psychosocial stressor on declarative memory [49] and on visuospatial abilities [50]. Other studies demonstrate that the neuroendocrine reproductive axis (measured by T levels) and stress axis (measured by C levels) interact to regulate dominance. For example, Mehta and Joseph [22] showed that T was positively related to dominance but only in individuals with low C. This effect was especially likely to occur after social threat.

A recent meta-analysis [51] investigated the associations between personality traits and neuroendocrinology, and found a very weak relationship between T and consciousness. Other effects regarding T and C and other personality traits were null or small. The authors suggested that basal levels of T and C (as measured in the present study), as opposed to context-specific measurements (times of competition or stress), may only partially be related to processes underlying the association between neuroendocrinology and personality. Future studies should, therefore, further examine the role of biology in procrastination in specific contexts that elicit neuroendocrine reactivity.

The present study has some limitations. First, the study is correlational, and, thus, causal effects between variables are not possible. A few studies manipulated anxiety to test how it can impact procrastination. For example, Bui [42] revealed that trait procrastination moderated the relation between anxiety and procrastination behavior: anxiety was manipulated through evaluation threat, while procrastination behavior was measured by the time when participants submitted their assignments. Results showed no main effect between anxiety and procrastination behavior. However, there was a significant interaction between anxiety and trait procrastination on procrastination behavior: under higher anxiety levels, procrastinators delayed more than non-procrastinators, while, under lower anxiety levels, non-procrastinators delayed more than procrastinators. The authors suggested that anxiety may potentially have the beneficial effect of reducing procrastination under certain conditions. Xu and colleagues [52] conducted experiments to investigate the impact of state anxiety, manipulated through a mental addition calculation, on procrastination. They found that participants with high state anxiety procrastinated less than participants with low state anxiety, thus, supporting the self-regulatory theory that negative emotions increase goal-directed behavior, thus, reducing procrastination. In contrast to Bui’s [42] findings, Xu and colleagues [52] did not find trait anxiety to have a moderating effect. Based on the current results, future experimental studies might shed light on the causal relationship between anxiety and procrastination by investigating biological moderating effects. Furthermore, it is important to examine previous studies’ suggestion that the relationship between anxiety and procrastination could be bidirectional.

Second, C levels were measured with regard to stress or other conditions connected to negative emotions through the measurement of basal levels or through C reactivity to induced stress. The present study did not detect a moderating effect of C on the relationship between anxiety and procrastination. Future studies should examine the linkage between these variables under various conditions, such as psychosocial or physical stress.

Finally, given previous inconsistent findings regarding the relationship between anxiety and procrastination, there is a need to develop a broader interactive model including both psychological and biological variables. Furthermore, apart from personality traits, other psychological variables explaining the phenomena of procrastination should be incorporated. Zhao and colleagues [15] suggested that, according to the self-regulation theory, there is a need to assess explicitly self-monitoring skills and observe behavioral changes through experimental design.

## 5. Conclusions

This is, to the best of our knowledge, the first study addressing the moderating role of neuroendocrinology on the relationship between anxiety and procrastination. We found that T moderates the association between state anxiety and procrastination: lower T levels strengthen the association, while under higher T levels, the association is diminished. It is possible that higher T levels have an anxiety-buffering function—a suggestion that should be further explored along with other social behaviors and academic outcomes.

## Figures and Tables

**Figure 1 behavsci-13-00204-f001:**
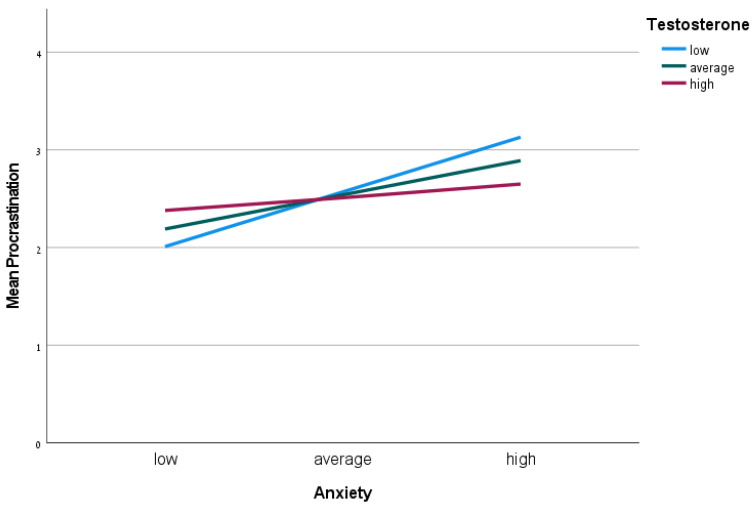
T×anxiety interaction for procrastination. A significant positive relationship between anxiety and procrastination (mean procrastination refers to the average score on the General Procrastination Scale [GPS]) is only found at low (blue line) and average (red line) testosterone levels. Note: Plotted points represent conditional low, average, and high values (±1 SDs) of testosterone and state anxiety.

**Table 1 behavsci-13-00204-t001:** Means (standard deviations), F, and p values for group differences in procrastination, state anxiety, and biomarkers (T, C).

	Men (N = 29)	OC (N = 32)	LP (N = 27)	F
Procrastination	2.36 (0.72)	2.35 (0.69)	2.54 (0.80)	0.60
State anxiety	29.59 (8.52)	33.19 (9.52)	35.67 (11.29)	2.74
T (nmol/L)	106.28 (48.77)	19.86 (9.73)	33.78 (22.74)	65.28 ***
C (nmol/L)	4.70 (2.80)	4.78 (2.95)	5.27 (4.27)	0.24

**** p* < 0.001 Note: T = testosterone; C = cortisol.

**Table 2 behavsci-13-00204-t002:** Correlations between hormones, procrastination, and state anxiety.

	1	2	3
Procrastination			
2. State anxiety	*** 0.40		
3. T	−0.00	−0.03	
4. C	−0.07	−0.10	0.17

*** *p* < 0.001 Note: T = testosterone; C = cortisol. 1 = procrastination; 2 = state anxiety; 3 = testosterone.

## Data Availability

The datasets used during the present study are available from the corresponding author on reasonable request.

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
