# Peer review of "State Anxiety and Procrastination: The Moderating Role of Neuroendocrine Factors"

_behavsci, 2023, doi:10.3390/bs13030204_

Round 1

Author Response

  • Authors need to write what software did they use for their test

We added details of the software version in the Methods (p. 4).

  • It seems you should have done ordinal regression rather than multiple regression because your dependent variable (procrastination) was measured on an ordinal scale.

The dependent variable (procrastination) was indeed measured on an ordinal scale; however, we adopted the approach that parametric statistics are robust with respect to violations of assumptions such as the use of ordinal scale (Norman, 2010). Furthermore, we followed studies using these statistical analyses (e.g., Sirois, 2004).

Norman, G. (2010). Likert scales, levels of measurement and the “laws” of statistics. Advances in health sciences education15, 625-632.‏

Sirois, F. M. (2004). Procrastination and counterfactual thinking: Avoiding what might have been. British journal of social psychology43(2), 269-286.‏

  • It is better to show the real number of p values rather than showing that p is smaller than 0.05. Line 211, for example, “p = 0.000” is incorrect. if the p-value is very small you can write that p is smaller than 0.001.

We changed the number of p values in the Results section (p.4-5).

  • The authors need to define what is the meaning of Low, average, and high Testosterone

We defined the levels of testosterone (p.4-6).

  • Table 2: the table legend must explain the table clearly and thoroughly. For example, correlations of what? what are 1, 2, and 3?

We added a thorough explanation in Table 2 (p.5).

  • Anxiety level should be defined. For example, what factors did you consider for a high level of anxiety

We defined the anxiety levels (p. 4).

  • Figure 1 needs to be improved. first, the figure legend should explain more details.

second: it needs to explain what the mean procrastination number (1,2,3) is. third: the digits after dots should be removed from the y-axis and their size and fonts should be matched to the passage. The authors need to define what is the meaning of Low, average, and high Testosterone in Figure 1!

We improved Figure 1 according to the reviewer’s comments. We elaborated the description of the figure, explained the procrastination scale, removed the decimals on the y-axis, and defined the meaning of T levels (p. 6).

  • Line 240: "Endocrine factors may serve as modulators of psychological outcomes" It is not a proper conclusion. Authors need to explain how testosterone serves as a modulator! Also, the authors need to explain why lower T levels strengthened the association between anxiety and procrastination while higher T levels induce no significant effect.

We rewrote the conclusion and provided an explanation of the findings regarding the moderating role of T levels on the association between anxiety and procrastination (p. 6).

Reviewer 2 Report

In this interesting paper by Barel and coworkers, the moderating role of neuroendocrine factors, i.e., testosterone (T) and cortisol, in the relationship between state anxiety and tendency to procrastination has been investigated in a sample consisting of 88 undergraduate students. The manuscript is well-written; nevertheless, the “good pen” hides some important gaps.

-Why did the authors employ the STAI-S? This questionnaire, as correctly underlined within the Methods section, provides a point estimate of anxiety experiences. It would make sense to attempt to induce anxiety experimentally, e.g., concerning a deadline, and in the context of such a simulation administer both questionnaires. In my opinion, the STAI-Trait should have to be implemented.

-I suggest reporting additional details regarding the psychometric properties and available norms of the STAI-Y.

For reference purposes, please refer to the following:

Bergua V et al. The STAI-Y trait scale: psychometric properties and normative data from a large population-based study of elderly people. Int Psychogeriatr. 2012; doi:10.1017/S1041610212000300

Ilardi CR et al. Psychometric properties of the STAI-Y scales and normative data in an Italian elderly population. Aging Clin Exp Res. 2021; doi:10.1007/s40520-021-01815-0

Potvin O et al. Norms and associated factors of the STAI-Y State anxiety inventory in older adults: results from the PAQUID study. Int Psychogeriatr. 2011; doi:10.1017/S1041610210002358

-The findings of this study are the results of a statistical artifact!!! The authors found that the lower the T levels, the stronger the association between anxiety and procrastination; conversely, this relationship would expire at higher T levels. It is obvious that T levels are higher in the male than in the female group. Therefore, results should be interpreted in the context of gender differences: the association between anxiety and procrastination exists in females, but not in males. In other words, the same results would have been obtained by performing two “simple” bivariate correlation analyses in the female and male subgroups. As for the former, the authors would find a significant and strong correlation between anxiety and procrastination; as for the latter, they would find, instead, no relationship. The authors are kindly requested to enter the dummy variable “sex” in the regression model as a covariate. Alternatively, they should run two separate regression analyses, one for males and one for females. In this case, the male subgroup should be at least doubled.

-Authors need to report existing data on gender differences on this topic and discuss their results with this in mind.

Author Response

  • Why did the authors employ the STAI-S? This questionnaire, as correctly underlined within the Methods section, provides a point estimate of anxiety experiences. It would make sense to attempt to induce anxiety experimentally, e.g., concerning a deadline, and in the context of such a simulation administer both questionnaires. In my opinion, the STAI-Trait should have to be implemented.

We provided an explanation of the use of STAI-S in the current study (p. 3).

  • I suggest reporting additional details regarding the psychometric properties and available norms of the STAI-Y.

For reference purposes, please refer to the following:

Bergua V et al. The STAI-Y trait scale: psychometric properties and normative data from a large population-based study of elderly people. Int Psychogeriatr. 2012; doi:10.1017/S1041610212000300

Ilardi CR et al. Psychometric properties of the STAI-Y scales and normative data in an Italian elderly population. Aging Clin Exp Res. 2021; doi:10.1007/s40520-021-01815-0

Potvin O et al. Norms and associated factors of the STAI-Y State anxiety inventory in older adults: results from the PAQUID study. Int Psychogeriatr. 2011; doi:10.1017/S1041610210002358

We provided additional details regarding the psychometric properties and available norms of the STAI-Y based on the references suggested by the reviewer (p. 3).

  • The findings of this study are the results of a statistical artifact!!! The authors found that the lower the T levels, the stronger the association between anxiety and procrastination; conversely, this relationship would expire at higher T levels. It is obvious that T levels are higher in the male than in the female group. Therefore, results should be interpreted in the context of gender differences: the association between anxiety and procrastination exists in females, but not in males. In other words, the same results would have been obtained by performing two “simple” bivariate correlation analyses in the female and male subgroups. As for the former, the authors would find a significant and strong correlation between anxiety and procrastination; as for the latter, they would find, instead, no relationship. The authors are kindly requested to enter the dummy variable “sex” in the regression model as a covariate. Alternatively, they should run two separate regression analyses, one for males and one for females. In this case, the male subgroup should be at least doubled.

In addition to standardizing T levels within the groups, we followed the reviewer’s suggestion and entered sex as a covariate in the regression (p. 5).

  • Authors need to report existing data on gender differences on this topic and discuss their results with this in mind.

We addressed the topic of sex differences in the Discussion section (p. 7).

Reviewer 3 Report

1. The average value of procrastination according to table 1 was 2.36-2.54. It is not clear how this correlates with the scale of the questionnaire. According to the received data, what number of volunteers have more / less points and, accordingly, can they be divided into subgroups according to the level of procrastination?

2. Also, according to the level of anxiety, the assessment on the scale is not clear. Add a description of the scale in the Materials and Methods section and comment on the homogeneity/heterogeneity of the sample.

3. Comparison of testosterone levels in groups of men and women is predictably skewed towards higher levels in men. It seems to me that correlation analysis should be carried out separately in the group of men and the group of women. In this case, there may be additional results.

4. There is no discussion of the results, taking into account the use of oral contraceptives. Why was this group included in the study?

Author Response

  • The average value of procrastination according to table 1 was 2.36-2.54. It is not clear how this correlates with the scale of the questionnaire. According to the received data, what number of volunteers have more / less points and, accordingly, can they be divided into subgroups according to the level of procrastination?

The average of procrastination conforms with previous findings (e.g., Lyons et al. 2014). Furthermore, study groups did not significantly differ in their levels of procrastination (p. 4).

  • Also, according to the level of anxiety, the assessment on the scale is not clear. Add a description of the scale in the Materials and Methods section and comment on the homogeneity/heterogeneity of the sample.

We elaborated the description of the scale and verified that the variance conforms with previous studies (e.g., Ilardi et al., 2021) (p. 3).

Ilardi, C. R., Gamboz, N., Iavarone, A., Chieffi, S., & Brandimonte, M. A. (2021). Psychometric properties of the STAI-Y scales and normative data in an Italian elderly population. Aging clinical and experimental research, 1-8.‏

  • Comparison of testosterone levels in groups of men and women is predictably skewed towards higher levels in men. It seems to me that correlation analysis should be carried out separately in the group of men and the group of women. In this case, there may be additional results.

In addition to standardizing T levels within the groups, we followed the reviewer’s suggestion and entered sex as a covariate in the regression (p. 5)

  • There is no discussion of the results, taking into account the use of oral contraceptives. Why was this group included in the study?

We addressed the inclusion of the oral contraceptives (OC) group (p. 3).